# Nanometric Mechanical Behavior of Electrospun Membranes Loaded with Magnetic Nanoparticles

**DOI:** 10.3390/nano13071252

**Published:** 2023-04-01

**Authors:** Raffaele Longo, Luigi Vertuccio, Vito Speranza, Roberto Pantani, Marialuigia Raimondo, Elisa Calabrese, Liberata Guadagno

**Affiliations:** 1Department of Industrial Engineering, University of Salerno, Via Giovanni Paolo II 132, 84084 Fisciano, Italy; rlongo@unisa.it (R.L.); vsperanza@unisa.it (V.S.); rpantani@unisa.it (R.P.); mraimondo@unisa.it (M.R.); elicalabrese@unisa.it (E.C.); 2Department of Engineering, University of Campania “Luigi Vanvitelli”, Via Roma 29, 81031 Aversa, Italy; luigi.vertuccio@unicampania.it

**Keywords:** electrospinning, nanocomposite membrane, magnetite nanoparticles, nanometric mechanical properties

## Abstract

This work analyzes on nanoscale spatial domains the mechanical features of electrospun membranes of Polycaprolactone (PCL) loaded with Functionalized Magnetite Nanoparticles (FMNs) produced via an electrospinning process. Thermal and structural analyses demonstrate that FMNs affect the PCL crystallinity and its melting temperature. HarmoniX-Atomic Force Microscopy (H-AFM), a modality suitable to map the elastic modulus on nanometric domains of the sample surface, evidences that the FMNs affect the local mechanical properties of the membranes. The mechanical modulus increases when the tip reveals the magnetite nanoparticles. That allows accurate mapping of the FMNs distribution along the nanofibers mat through the analysis of a mechanical parameter. Local mechanical modulus values are also affected by the crystallinity degree of PCL influenced by the filler content. The crystallinity increases for a low filler percentage (<5 wt.%), while, higher magnetite amounts tend to hinder the crystallization of the polymer, which manifests a lower crystallinity. H-AFM analysis confirms this trend, showing that the distribution of local mechanical values is a function of the filler amount and crystallinity of the fibers hosting the filler. The bulk mechanical properties of the membranes, evaluated through tensile tests, are strictly related to the nanometric features of the complex nanocomposite system.

## 1. Introduction

In the last two decades, nanotechnologies have been proposed in many fields. Their potentialities are extremely promising for a wide range of applications [1,2,3,4]. For example, carbonaceous nanofillers such as graphene-based nanoparticles, carbon nanotubes, fullerenes, etc., are extensively studied because of their peculiar properties, e.g., electrical conductivity and, thermal conductivity, that were previously limited to metals, attracting great attention from the industries in which these properties are strongly required [5,6,7]. Similarly, metallic nanoparticles (based on Au, Ag, etc.) are now extremely interesting for their peculiar optical properties and intrinsic bioactivity [8,9]. Moreover, most of these nanoparticles (from magnetic nanoparticles to metallic) are reactive to external stimuli, and, for this reason, they are suitable for the design of smart devices that require the ability to sensitively change a property of the device during its application (e. g. increasing the temperature, increasing the release kinetics of a drug, etc.) [10]. In light of these results, it is clear why a big focus of material scientists is the exploration of the transfer of the properties of the metallic nanoparticles to polymeric matrices. The production of nanocomposite polymeric systems is one of the simplest ways to effectively vary the properties of bulk material [11]. Moreover, thanks to their nano-dimensions, the fillers have a very high exposed surface area, being so effective even at very low percentages, and also are suitable to be included in materials characterized by nanometric textures [12].

Typically, polymeric materials with this exciting peculiarity of being composed of nanostructured fiber mats are those obtained by electrospinning processes. Electrospun membranes are of great interest from an applicative point of view. They can be designed for application as filtering devices [13,14,15,16,17,18], in biomedical applications and tissue regeneration [19,20,21,22], as promoters of antibacterial properties [23,24,25,26] or as modifiers, in the form of thermoplastic veils, for improving interlaminar toughness of fiber-reinforced epoxy composites, and for dispersing in them nanofillers [27,28]. 

In the production of biomaterials for tissue engineering, for example, it is generally desirable to have materials with the same texture of the primary tissue. 

The electrospinning process allows the efficient and easy production of nanofibrous membranes, which well-mimic the morphology of human tissue or specific biological tissues. Together with the morphological feature, the possibility of tailoring and controlling the mechanical properties at the macroscopic scale but also at the nanoscopic level in the form of mapping of specific mechanical parameters is of great relevance.

The use of nanotechnologies in electrospun membranes is looked at as one of the most promising pathways to tailor mechanical and functional characteristics, for ex. for producing highly flexible, microporous nanomaterials that convincingly mimic the features of the extracellular matrix and human tissue in general. Moreover, the possibility to easily load different types of fillers in the electrospun matrix for all types of applications is attracting great attention because it represents a powerful method to direct the material toward a myriad of diversified performances, including smart functions [29,30,31,32,33]. Moreover, by properly functionalizing the nanoparticle surface, it is possible to control its hydrophilic/hydrophobic behavior (a crucial parameter for the dispersion in water and for cell uptake) [34,35,36]. However, the coating agents are generally polymers, polysaccharides, or active agents useful for their application in the biological environment [10,37,38,39,40]. 

Different research papers focused their attention on the control of the smart functionalities in these types of systems. For example, in the case of magnetic nanoparticles, the electrospun mat has demonstrated sensitivity to an external magnetic field, and capability to increase the temperature locally thanks to the application of an alternate magnetic field [10,41]. However, few research papers have deeply analyzed how the inclusion of electromagnetically responsive nanoparticles affects the morphological and local mechanical features and the local surface properties of the electrospun membranes.

In this paper, membranes loaded with magnetite nanoparticles compatibilized with the human environment were successfully embedded in the mat fibers following the procedure already reported in the literature [31]. These systems have proven efficacy in treating solid tumors, from cervix uterine to melanoma. Generally, the patients who suffer from these diseases should undergo the surgical remotion of the cancerous zone. After this step, chemotherapy treatment is provided to the patient to avoid any potential regrowth of the dangerous tissue and to treat the tissue that cannot be removed during surgery. In this scenario, topical chemotherapy treatment must be directly applied in a wet environment, covering the damaged tissue and possibly, providing the required antitumoral treatment. For this reason, it is necessary to analyze the structural characteristics and surface properties along the fiber surface. Recent relevant studies have proven that the evaporation of the solvent and the stresses provided during the electrospinning process intrinsically cause the development of nanofiber with a higher amount of voids and amorphous in the outer part of the nanofibers, having consequently less performing mechanical properties in the outer shell compared to the core [42]. Gòmez-Pachòn et al. [43] have noticed that during the electrospinning process of poly(lactic acid), it is possible to produce an amorphous and crystalline zone with lamellar-like structures. In fact, crystallinity plays a key role in these types of systems; recent research has proven that cell proliferation is strictly affected by the degree of crystallinity and types of cell lines [44,45]. Recent research papers have analyzed the nanometric mechanical properties of electrospun membranes. One of the most promising techniques is H-AFM, which allows a quantitative evaluation of the local mechanical modulus to be obtained. The performed studies are mainly focused on applying these systems in the biomedical field [46,47] and understanding how the local mechanical modulus depends on the polymer chemistry [48,49] and the structure of the nanofiber [42]. Based on these studies, to the authors’ knowledge, this is the first time that the nanometric mechanical properties were evaluated via H-AFM on electrospun nanocomposite membranes. Through this investigation, it was found that the inclusion of magnetite in electrospun membranes affects the crystallinity of the material and the distribution of the local mechanical properties along the mat fibers. Bulk mechanical tests have been compared with the results obtained by H-AFM. 

## 2. Materials and Methods

### 2.1. Materials 

PCL, with the CAS number 24980-41-4, was supplied by Perstorp (Malmö, Sweden) for the membrane preparation. It is a linear polyester with a high molecular weight (about 80,000) that was provided in pellet form (granules with an appr-oximate diameter of 3 mm). Aldrich Chemical Co. (Milwaukee, WI, U.S.) provided the iron (III) acetylacetonate (Fe(acac)_3_) (97%), 1,2-hexadecanediol (90%), oleic acid (OA), citric acid (CA), benzyl ether (90%), ethanol, hexane, and acetone. All of the chemicals used in the testing were of analytical grade. Fe_3_O_4_ magnetic nanoparticles (Fe_3_O_4_@OA) were made utilizing an experimental method that has previously been documented in the literature [50,51]. Fe_3_O_4_@OA was functionalized by altering the produced nanoparticles via ligand exchange with citric acid (CA) at room temperature, as described by Guadagno et al. [31]. The final functionalized nanoparticles are here labelled F-Fe_3_O_4_.

### 2.2. Electrospinning Procedure

The procedure followed to obtain the nanofibrous mats of PCL loaded with magnetite nanoparticles had been recently optimized by the research group. [31] PCL was dissolved in hot acetone (14% wt./wt.) and stirred for about two hours. To create homogeneous and stable solutions, nanoparticles were introduced to the system, and the solutions were then ultrasonically processed. F-Fe_3_O_4_ concentrations in PCL were calculated at 2, 5, and 10 wt.%, with respect to PCL concentrations. Climate-controlled electrospinning apparatus was used to electrospin the functionalized Fe_3_O_4_ nanoparticle-containing solutions (EC-CLI by IME Technologies, Spaarpot 147, 5667 KV, Geldrop, The Netherlands). Each combination was put into a stainless-steel syringe (3 mL) with a 0.8 mm needle. The needle had a high-voltage connection to a power source.

The counter electrode was a grounded piece of aluminium foil positioned 30 cm away from the spinneret. On the aluminium foil, PCL fibres in the shape of continuous filaments were gathered to form a fibrous membrane. The electrospinning process parameters were suitably set to prevent bead development and, hence, to manufacture mats of fibrous PCL made up of continuous individual fibrils, as homogeneous as possible in diameter along the length.

During the membrane fabrication process, the parameters were optimized through a process of trial and error. The voltage for electrospinning was set at 25 kV (charge applied to the collector at −4 kV and charge applied to the needle at 21 kV), feeding rate at 4.0 mL/h, the temperature at 25 °C, and relative humidity at 35%. The produced membranes were placed in a vacuum oven for 24 h at room temperature to remove any solvent traces. The membranes made were labelled “PCL_FillerPercentage_F-Fe_3_O_4_”.

### 2.3. Thermal Analysis

The thermal stability of the nanocomposite membranes was examined using thermogravimetric investigations (TGA-DTGA) using a Mettler Toledo TGA/STDA851e in flowing air at a 10 K/min heating rate.

Setting a heating rate of 10 °C per minute, differential scanning calorimetry (DSC) was carried out using a Mettler Toledo DSC 822e in an N_2_ environment at a flow rate of 50 mL/min.

By considering the enthalpy of fusion (Δ*H_m_*) and comparing it to the enthalpy of fusion of 100% PCL (Δ*H*_100%*m*_), 136.1 J/g, the crystallinity evaluation was evaluated through Equation (1):(1)$$X_{c}=\frac{\Delta H_{m}}{\left(1-\alpha \right)\times \Delta H_{100{\%}_{m}}}$$
where *α* is the mass fraction of magnetite in the nanocomposite. 

### 2.4. Structural Investigation

Structural characterization of the electrospun membranes was carried out by employing the diffractometer Bruker D8 Advance diffractometer (Bruker Corporation, Billerica, MA, USA) operating at 35 kV and 40 mA. The analysis was conducted in the 2θ range of 10–30 degrees. The data were analysed with the same methodology reported by Naddeo et al. [52]. The crystallinity of the samples was obtained according to Sownthari et al. [53] by deconvoluting the spectra [54,55] and considering the crystalline and the amorphous area [56] under the diffractometric curve profile, in accordance with procedure already reported in literature [29].

### 2.5. Nanometric Mechanical Mapping

Generally, to study electrospun membranes with functional nanoparticles, the physicochemical properties of the embedded nanoparticles in regulating nano-bio interactions are well-recognized aspects, as well as the effects of the size, shape, and surface charge of nanoparticles on their biological performances. All these aspects were extensively investigated in the literature. The role of nanoparticle mechanical properties in drug delivery has only been recognized recently and remains the least explored. Recently an interesting article review provided an overview of the impacts of nanoparticle mechanical properties on cancer drug delivery [57]. The article also deals with the current methods for fabricating nanoparticles with tunable mechanical properties. The different nanoparticles’ mechanical characteristics strongly affect mechanisms that control the complicated nano-bio interactions at the cellular, tissue, and organ levels. Furthermore, in many performed studies, a dependence of the size and shape of nanoparticles on antitumoral activity has already been observed. The size and shape of nanoparticle or nanoparticle domains are expected to be strictly related to their mechanical properties. As described in previous papers, the antitumor activity of the developed membranes is due to the presence of magnetic nanoparticles [31]. It is evident that the local mechanical properties, resulting from the rigid domains of magnetic nanoparticles are properties to be thoroughly investigated. To analyse the morphology of the samples and correlate it to the local mechanical properties, atomic force microscopy (AFM) with the HarmoniX tool was used. By utilizing the interactions between the tip and the sample surface, H-AFM is a technique that can generate a topographic and mechanical map of the sample surface [58]. Accurate data on the morphology and local elastic modulus of the sample’s surface can be obtained by measuring the cantilever’s deflection using a laser. With a nominal tip radius of 10 nm, the HMX-10 probe was used to conduct AFM measurements. 

This probe makes mapping nanometric mechanical characteristics possible between 10 MPa and 10 GPa. The elastic modulus was measured with an accuracy of 10 MPa and positioned on the sample surface with a precision of 10 nm. The Derjaguin-Muller-Toporov (DMT) model was used to process the height and local mechanical data using the procedure reported previously in the literature [59,60], which were processed along the fibres using NanoScope Analysis 1.40 Software (Bruker Corporation, Billerica, MA, USA) and via OriginPro 2018 64-bit software (OriginLab Corporation, Northampton, MA, USA). The modulus values assessed were taken along the nanofiber’s spine, in line with earlier research reports. Because of the curvature of the fibres, the AFM tip really has varied contact areas with the nanofiber’s surface, resulting in variations in the applied stress [48,61].

### 2.6. Bulk Mechanical Properties 

Mechanical characterization was carried out using Dual Column Tabletop Testing Systems (INSTRON, series 5967-INSTRON, Norwood, MA, USA) in tensile tests. Five samples of each membrane, each measuring 1 cm in width by 5 cm in length and 0.1 mm in thickness, were tested at room temperature. Tensile tests were performed by elevating the beam at 10 mm per minute while collecting information on the force and distance travelled. The data were then detected as stress-strain curves.

## 3. Results and Discussion

### 3.1. Thermal and Structural Analysis

TGA measurements were carried out on the obtained membranes. The results, reported in Figure 1, show that the PCL thermal degradation is anticipated by the inclusion of magnetite inside the matrix. This behaviour is probably due to the fact that the functionalized Fe_3_O_4_ nanoparticles are responsible for the random pyrolysis of PCL chains and accelerate the thermal degradation of PCL; a phenomenon already found in the literature for magnetite-based nanoparticles [62]. As reported in Table 1, the 5% weight loss is anticipated from 370 °C to around 330 °C by the inclusion of functionalized magnetite. However, this anticipation is not a problem for the system’s applicability here proposed since the study aims to explore the properties of these membranes for biomedical applications. As expected, increasing the magnetite content increases the residual ratios at 700 °C. However, as also reported in other recent papers in the literature, the residual quantity after the TGA does not perfectly match the theoretical amount of magnetite [63,64]. However, it is worth noting two aspects: (i) the degradation of the functionalized magnetite occurs (in particular, the degradation of the citric acid), causing more than 10% weight loss of the magnetite [31]; (ii) the magnetite anticipates the degradation of the PCL sensitively, suggesting that the type of degradation and, consequently, the type of residue may change between the PCL membranes loaded with magnetite and the unloaded one [62]. 

DSC analyses evidence that these types of systems are thermally stable (no transitions) around human body temperature. In fact, PCL, which is characterized by a low melting point (around 60 °C), tends to increase the melting temperature by increasing the F-Fe_3_O_4_ amount. The DSC graph is reported in Figure 2.

The inclusion of the filler in the polymeric matrix plays a relevant role. It causes an increase in the melting temperature, since it causes an increase in the crystal size in the matrix, as detectable by the narrowing of the melting endotherm profile, which becomes progressively narrower as the amount of magnetite increases with a simultaneous shift at a higher temperature with respect to the PCL alone. The crystallinity of the PCL, evaluated as the difference of melting enthalpy, is compared to that of a theoretical 100% PCL (136.1 g/mol). This analysis (reported in Table 2) proved that the inclusion of magnetite enhances the crystallinity of PCL, increasing from 39% (for 0 wt.% of F-Fe_3_O_4_) to 64% (for 5 wt.% of F-Fe_3_O_4_). In a PCL membrane loaded with 10 wt.% of F-Fe_3_O_4_, the large amount of magnetite hinders the crystallization process of the polymer, determining a decrease in the crystallinity degree to 43%.

All these data were confirmed by XRD analysis. The procedure to evaluate the crystallinity degree (X_c_) and the crystallite coherence lengths perpendicular to reflection planes 110 (D110), 200 (D200), and 111 (D111) for the PCL component of the membranes is described in Appendix A. 

Figure 3 shows the diffractograms of the developed samples. A higher crystallinity for the PCL_5%F-Fe_3_O_4_ was detected by evaluating the XRD spectra. Moreover, by applying the Scherrer Equation to the various typical peaks of the orthorhombic structure of PCL [31,45] ((110) at 21.5°, (111) at 22.0°, (200) at 23.8° of 2θ), it is evident that the crystallite size increases when the filler content increases (see Table 3), and this phenomenon well explains the increase in the melting temperature observed by the DSC analysis.

The evaluation of F-Fe_3_O_4_ distribution along the fibers of the membranes was performed using Energy Dispersive X-ray Analysis (EDX) investigation. The results shown in Appendix A evidence a good dispersion. 

### 3.2. Nanometric H-AFM Analysis and Mechanical Properties

The local mechanical characteristics of the nanocomposites were evaluated with an innovative approach, using the H-AFM, through which it is possible to obtain an accurate mechanical mapping of the nanofiber surfaces. The height and DMT modulus images are shown in Figure 4. As evident, the height image shows that the magnetite nanoparticles are included in the nanofibers. Moreover, the DMT modulus images show that the white dots (that correspond to higher mechanical modulus) are well distributed along the nanofibers. The formation of nanoparticle clusters slightly increases as the magnetite amount increases. Still, the adopted functionalization effectively spreads the magnetite nanoparticles in the PCL matrix homogeneously.

As described before, including magnetite nanoparticles in the matrix affects the crystallinity. This constatation explains the behaviour of the H-AFM results.

As reported in Figure 5, the DMT moduli increase by increasing the content of magnetite. However, by following the DMT profile along the fibres, it is possible to observe that passing from PCL with 0 wt.% of filler to PCL with 5 wt.% of filler, the mechanical values increase by increasing the magnetite content (since the crystallinity is higher). A comparison with the thermal results evidences that for PCL with 10%F-Fe_3_O_4_, with more perfect PCL crystals (and lower crystallinity degree), the presence of higher peaks is due to the presence of magnetite aggregates in the matrix together with the presence of more perfect crystals. It is also possible to see that the aggregates can reach an extension of the aggregate domains up to 550–750 nm.

By graphing the distribution of values obtained for the various membranes, as reported in Figure 6, it is possible to observe that the average mechanical modulus increases by increasing the magnetite content. However, as noticed in Figure 6 and in Table 3, the crystallinity and the magnetite content are key parameters that affect the local mechanical modulus distribution. For this reason, the PCL_10%F-Fe_3_O_4_ system presents overall an average mechanical modulus higher than PCL_5%Fe_3_O_4_, and a wider distribution, because of a more significant number of magnetite aggregates. This aspect was also considered by evaluating the distribution’s asymmetry. The skewness of the various distributions was evaluated (reported in Table 4) using Equation (2).
(2)$$\mathrm{Skewness}=\overset{N}{\underset{i=1}{\sum }}\frac{{\left(X_{i}-X^{*}\right)}^{3}}{\left(N-1\right)\times \sigma^{3}}\;\;$$

This parameter gives information about the asymmetry of the distribution of the local mechanical values. [65] It increases by increasing the filler amount, but the higher difference is for the 10 wt.% of F-Fe_3_O_4_. Generally, a distribution curve can be symmetrical if the skewness is between −0.5 and 0.5. If the skewness is above 0.5, it is considered asymmetrical. In this case, the asymmetry of the PCL_10%F-Fe_3_O_4_ curve is due to the crystallinity degree (that causes a decrease in the mode); in contrast, there is a higher quantity of nanoparticles and clusters, which causes an increase in the local mechanical values. For this reason, the distribution is asymmetrical and shows a strong dependence on the nanoparticle percentage. 

In synthesis, the mode is a more accurate statistical parameter to monitor the DMT modulus of the matrix, whereas the average is the statistical parameter that describes the properties of the whole nanocomposite. The local mechanical properties of the material were compared to the bulk mechanical properties in tensile mode, and the results are reported in Figure 7. The increase in Fe_3_O_4_ loading increases the material’s elasticity, so the local mechanical properties well explain the increase in bulk properties. The Young modulus of the PCL (4.58 MPa) slightly increases by increasing the magnetite content from 5 wt.% (6.40 MPa) to 10 wt.% (6.81 MPa), following the trend of the nanometric measurements. The elastic values measured at the nanoscale differ from the bulk values since the H-AFM evaluates the mechanical properties by tapping the sample (similar to the indentation), whereas the bulk tests are in tensile. It is worth noting that the strain at the break of the various membranes is not significantly affected by the inclusion of magnetite nanoparticles, that in fact is around 200% for all the membranes. It is probably due to the fact that the bulk properties of electrospun membranes also depend on other parameters, such as the fibre dimensions formed by the electrospinning process [66] or the fibre density in the mat.

It is worth noting that by performing a comparison between the elasticity modulus obtained on nanometric domains of the sample surface through H-AFM mode and that obtained from stress-strain curves of the bulk membrane, the values of the elastic modulus detected by H-AFM seem overestimated, even if the trend is the same applying the two different modes. Considering the quantitative difference in the modulus values obtained using the different modes, the following question arises spontaneously: how reliable is the calculation procedure through H-AFM if one wanted to use the data to get a quantitative map of the elastic modulus on the surface? To answer this question, we have to consider two aspects: one concerns the acquisition method through the H-AFM mode, the other concerns the heterogeneity of the material under observation. Although magnetic nanoparticles are functionalized to optimize their dispersion along the fibres, they are still responsible for the heterogeneity in the local mechanical response, as well detectable by DMT Modulus profiles along the fibre lengths in Figure 4. The elastic moduli of the samples detected using the H-AFM mode according to the DMT model [67,68] take into account the force–indentation curves following Equation (3)
(3)$$F_{L}\left(i\right)=\frac{4}{3}\times E^{*}\times \sqrt{R^{*}}\times i^{\frac{3}{2}}+F_{pull-off}$$
where *F_L_* is the load force, *E** is the reduced Young’s modulus, *E** = *E*/(1 − ν^2^), ν is the Poisson ratio, *R** is the reduced radius 1/*R** = 1/*R*_indenter_ + 1/*R*_surface_, *i* is the indentation depth, and *F_pull-off_* is the force at the point of pull-off of the AFM probe, or the pull-off force for short. The pull-off force can easily be found from the force–indentation curves.

By using the DMT model, the force–indentation is strongly affected by the heterogeneity of the sample. In particular, the indentation force on the magnetic nanoparticles is strongly affected by the rigidity of the magnetite domains. The average DMT modulus of the functionalized magnetite nanoparticle was found to be around 6 GPa. This value agrees with the value found in literature by the other authors for the magnetite hardness alone [69]. In summary, in the traditional tensile test, the elastic forces determined by the entanglement of the polymeric chains play a decisive role in conditioning the modulus values, whereas, for the modulus determined by DMT mode, the more relevant contribution results from the rigid domains of magnetic nanoparticles. 

Both methods of acquiring the mechanical properties are relevant because they allow the membranes to be designed from two different points of view. The one based on the tensile tests of the material allows considerations on the mechanical consistency of the material from the point of view of the macroscopic application, while the one relating to the local mechanical properties allows information on aspects related to bioreactivity (e.g., interaction with cells, biological tissues, etc.) to be obtained, as described in Section 2.5.

## 4. Conclusions

In this paper, the inclusion of functionalized magnetite nanoparticles in electrospun membranes and their effect on the nanometric properties of the material were explored. The inclusion of nanoparticles affects the thermal and structural properties of the material. Differential scanning calorimetry analyses reveal a strong variation of crystallinity by increasing the magnetite amount from 0 wt.% to 10 wt.%. This dependence is non-monotonous since, for a high amount of nanoparticles, the filler starts to hinder the crystallization of the polymer. The nanometric mechanical properties were explored via an innovative approach through H-AFM, which was used to track the local mechanical modulus along the nanofiber lengths. The results evidence that the nanometric mechanical properties of the samples are a function of the filler percentage and the crystallinity degree. The variation in the bulk mechanical properties is in accordance with the nanometric analysis, reporting an Increase of the Young modulus by increasing the filler percentage. 

## Figures and Tables

**Figure 1 nanomaterials-13-01252-f001:**
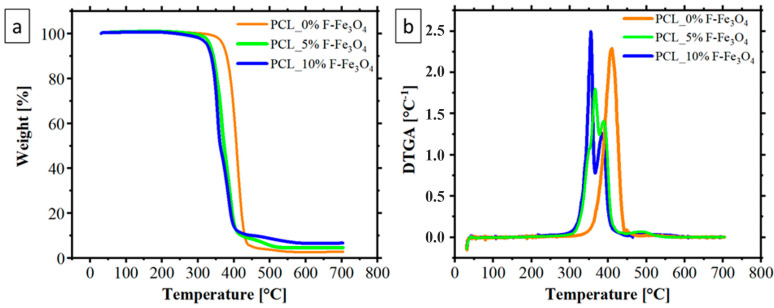
TGA (**a**) and DTGA (**b**) of PCL membranes loaded with F-Fe_3_O_4_ nanoparticles.

**Figure 2 nanomaterials-13-01252-f002:**
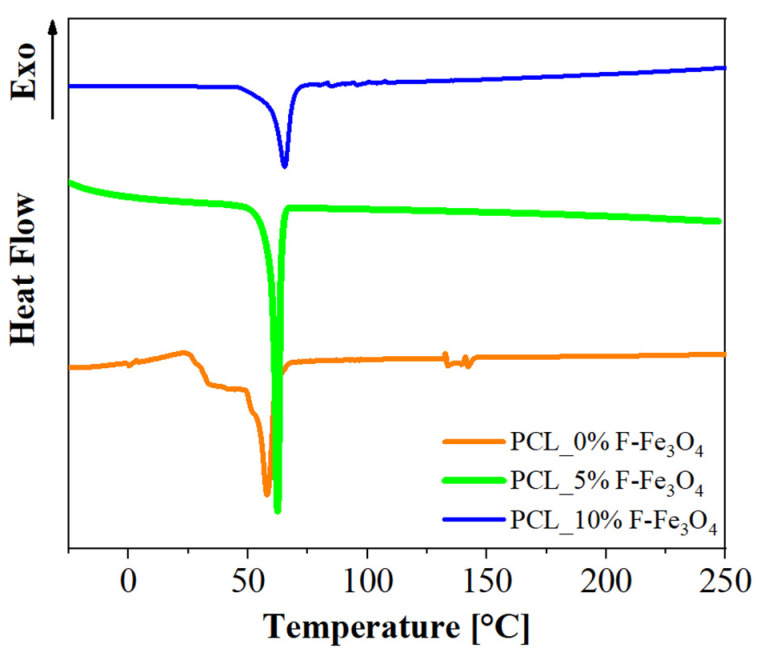
DSC curves of PCL membranes loaded with F-Fe_3_O_4_.

**Figure 3 nanomaterials-13-01252-f003:**
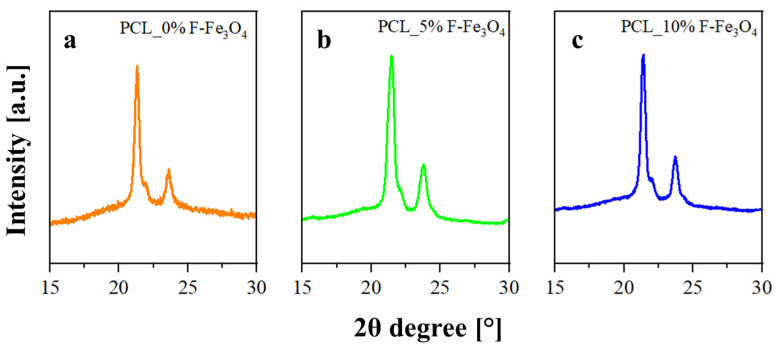
XRD spectra of: (**a**) PCL_0%F-Fe_3_O_4_; (**b**). PCL_5%F-Fe_3_O_4_; (**c**) PCL_10%F-Fe_3_O_4_;.

**Figure 4 nanomaterials-13-01252-f004:**
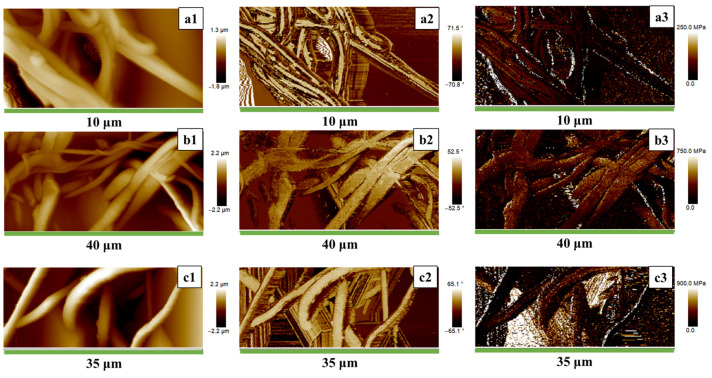
H-AFM images of PCL membranes loaded 0% (**a**), 5% (**b**) and 10% (**c**) of F-Fe_3_O_4_ nanoparticles. The images in the first column (**a1**,**b1**,**c1**) show the Height profile; the images in the second column (**a2**,**b2**,**c2**) show the Phase (**a2**,**b2**,**c2**); the images in the third column (**a3**,**b3**,**c3**) show the DMT Modulus.

**Figure 5 nanomaterials-13-01252-f005:**
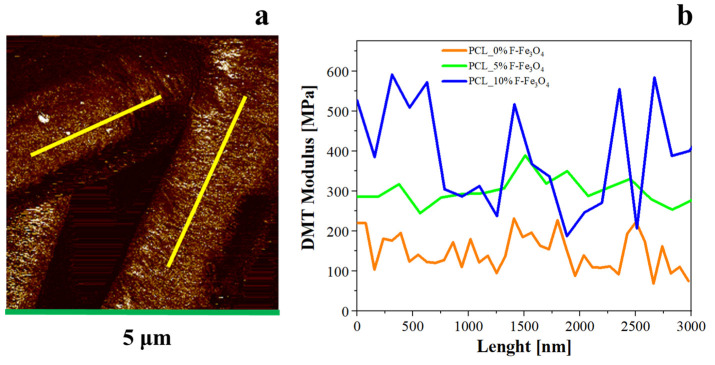
(**a**) H-AFM image of PCL membranes loaded with magnetite (5%). DMT Modulus profiles have been taken along the fibre length (see yellow lines). (**b**) Comparison of the DMT Modulus profiles along the fibre lengths for the analysed samples.

**Figure 6 nanomaterials-13-01252-f006:**
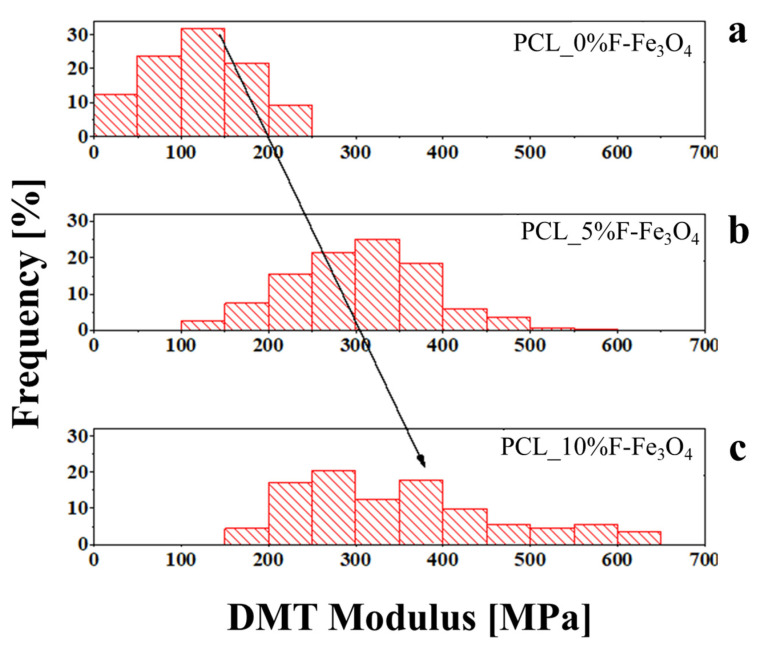
DMT frequency values of: (**a**) PCL_0%F-Fe_3_O_4_; (**b**) PCL-5%F-Fe_3_O_4_; (**c**) PCL-10%F-Fe_3_O_4_.

**Figure 7 nanomaterials-13-01252-f007:**
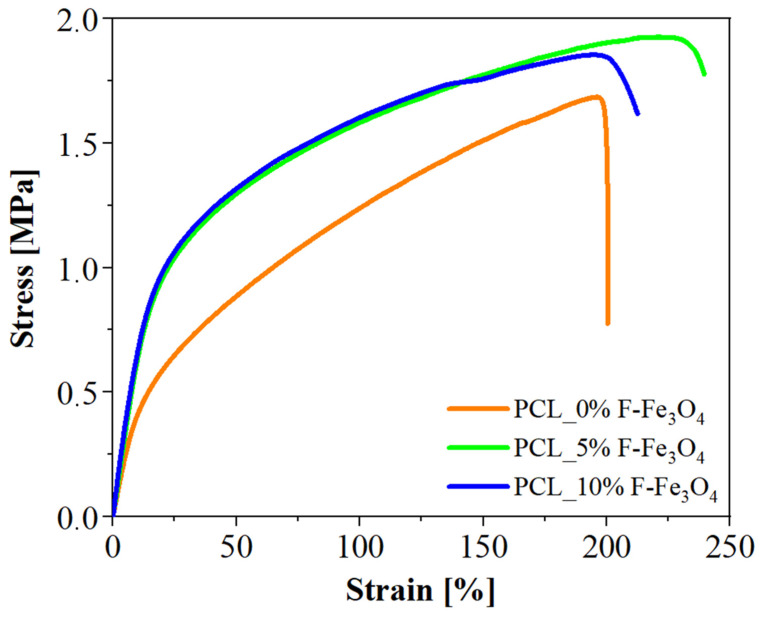
Stress-strain curves of Fe_3_O_4_ loaded membranes.

**Table 1 nanomaterials-13-01252-t001:** 5%, 50% weight loss and residuals at 700 °C for PCL membranes loaded with Functionalized Fe_3_O_4_ nanoparticles.

	T_loss,5%_ [°C]	T_loss,50%_ [°C]	Residuals at 700 °C
PCL_0%F-Fe_3_O_4_	370	407	2.68%
PCL_5%F-Fe_3_O_4_	333	373	4.53%
PCL_10%F-Fe_3_O_4_	325	362	6.59%

**Table 2 nanomaterials-13-01252-t002:** Melting Temperature and Crystallinity of PCL-loaded membranes.

Sample	Melting Temperature [°C]	Crystallinity [%]
PCL_0%F-Fe_3_O_4_	58.3	39.1
PCL_5%F-Fe_3_O_4_	62.2	64.5
PCL_10%F-Fe_3_O_4_	65.5	43.5

**Table 3 nanomaterials-13-01252-t003:** Crystallite coherence lengths perpendicular to reflection planes 110 (D110), 200 (D200), and 111 (D111) and crystallinity values of the unloaded PCL membrane and membranes loaded with 5% and 10% F-Fe_3_O_4_ nanoparticles.

Sample	D110 [Å]	D111 [Å]	D200 [Å]	Crystallinity [%]
PCL_0%F-Fe_3_O_4_	180	176	153	43.26
PCL_5%F-Fe_3_O_4_	184	213	155	60.11
PCL_10%F-Fe_3_O_4_	206	270	193	48.82

**Table 4 nanomaterials-13-01252-t004:** Statistical parameters of local mechanical value distribution.

Parameters	PCL	PCL_5%F-Fe_3_O_4_	PCL_10%F-Fe_3_O_4_
Average DMT Modulus [MPa]	120.7	303.8	352.2
Standard Deviation [MPa]	57.67	80.15	116.2
Mode [MPa]	125	325	275
Skewness	0.079	0.131	0.705

## Data Availability

Not applicable.

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
