# Peer review of "Nanometric Mechanical Behavior of Electrospun Membranes Loaded with Magnetic Nanoparticles"

_nanomaterials, 2023, doi:10.3390/nano13071252_

Round 1

Reviewer 1 Report

In this paper, the inclusion of functionalized magnetite nanoparticles in electrospun membranes was fabricated. The thermal, nanometric and bulk mechanical properties have been characterized and the results showed that the relevant properties have been enhanced. This manuscript is recommended to be publishable after addressing the following concerns.

1. Format:the format for reference citation is wrong, the reference number in main text should not be before the end of each sentence; Line 196, there is a format mistake.

2. The introduction part, some recent published papers about functionalization and AFM characterization of electrospun mats should be mentioned to enrich the significance and background of the present work. 

The nanometric electrostatic mapping of electrospun fibers through EFM should be cited: A flexible electrostatic nanogenerator and self-powered capacitive sensor based on electrospun polystyrene mats and graphene oxide films[J]. Nanotechnology 32 (2021) 405402 (10pp).

The recent report about loading nanofillers in the electrospun matrix to tailor mechanical properties and endow functional characteristics should be cited: Highly conductive, stretchable, durable, breathable electrodes based on electrospun polyurethane mats superficially decorated with carbon nanotubes for multifunctional wearable electronics[J]. Chemical Engineering Journal 451 (2023) 138549.

3. In the TGA test, the weight ratios of the residuals for differenct composite after thermal degradation should be given. Did these residuals are consistent with the added Fe3O4 weight ratios?

Reviewer 2 Report

The authors prepared nanocomposite materials of polycaprolactone (PCL) and magnetite nanoparticles and attempted to evaluate nanometric mechanical properties of them. Although the prepared materials might be useful, the manuscript suffers from vital problems. They are given below.

-Significance of the nanometric evaluation is unclear. When the elastic modulus results obtained by the nanometric and macroscopic evaluations were compared, just quantitative consistency was found. Usual macroscopic measurements of mechanical properties will be enough to discuss properties of the nanocomposite materials.

-The superiority of nanometric evaluation with AFM in the present study is restricted. To unveil distribution of magnetite nanoparticles in polymeric matrices, TEM observation would be more immediate and powerful method. The nanometric evaluation with AFM will work effectively for polymer-polymer composites such as polystyrene/PCL composites.

-“HarmoniX” is a commercial name of AFM measurement procedures and is not a common wording. Features of HarmoniX should be given briefly and proper preceding papers should be cited in the Introduction section.

-Line 236: The authors alleges that the XRD profiles agree with DSC results. Among the XRD profiles given in Fig. 3, however, the peaks in (b) seem to be least sharp. Re-examine the FWHM of the XRD reflection peaks.
